# Goal Representations for Instruction Following:
# A Semi-Supervised Language Interface to Control

**Vivek Myers,**[*] **Andre He,**[*] **Kuan Fang, Homer Walke, Philippe Hansen-Estruch,**
**Ching-An Cheng,**[†] **Mihai Jalobeanu,**[†] **Andrey Kolobov,**[†] **Anca Dragan, Sergey Levine**
University of California Berkeley, [†]Microsoft Research
{vmyers,andre.he,kuanfang,homer_walke,hansenpmeche}@berkeley.edu
{ChingAn.Cheng,mihaijal,akolobov}@microsoft.com
{anca,svlevine}@berkeley.edu

**Abstract:** Our goal is for robots to follow natural language instructions like "put the towel next to the microwave." But getting large amounts of labeled data, i.e. data that contains demonstrations of tasks labeled with the language instruction, is prohibitive. In contrast, obtaining policies that respond to image goals is much easier, because any autonomous trial or demonstration can be labeled in hindsight with its final state as the goal. In this work, we contribute a method that taps into joint image- and goal- conditioned policies with language using only a small amount of language data. Prior work has made progress on this using vision-language models or by jointly training language-goal-conditioned policies, but so far neither method has scaled effectively to real-world robot tasks without significant human annotation. Our method achieves robust performance in the real world by learning an embedding from the labeled data that aligns language not to the goal image, but rather to the desired *change* between the start and goal images that the instruction corresponds to. We then train a policy on this embedding: the policy benefits from all the unlabeled data, but the aligned embedding provides an *interface* for language to steer the policy. We show instruction following across a variety of manipulation tasks in different scenes, with generalization to language instructions outside of the labeled data. Videos and code for our approach can be found on our website: https://rail-berkeley.github.io/grif/

**Keywords:** Instruction Following, Representation Learning, Manipulation

## 1  Introduction

Natural language has the potential to be an easy-to-use and powerful form of task specification in robotics. To follow language instructions, a robot must understand human intent, ground its understanding in the state and action spaces, and solve the task by interacting with the environment. Training robots to do this is challenging, especially given that language-annotated data is limited. Existing deep learning approaches require large amounts of expensive human language-annotated demonstrations and are brittle on instructions outside the training data.

Visual goals (i.e., goal images), though less intuitive for humans, provide complementary benefits for task representation in policy learning. Goals benefit from being easy to ground since, as images, they can be directly compared with other states. More importantly, goal tasks provide additional supervision and enable learning from unstructured data through hindsight relabeling [1, 2, 3]. However, compared to language instructions, specifying visual goals is less practical for real-world applications, where users likely prefer to tell the robot what they want rather than having to show it.

---

[*]Equal contribution

7th Conference on Robot Learning (CoRL 2023), Atlanta, USA.

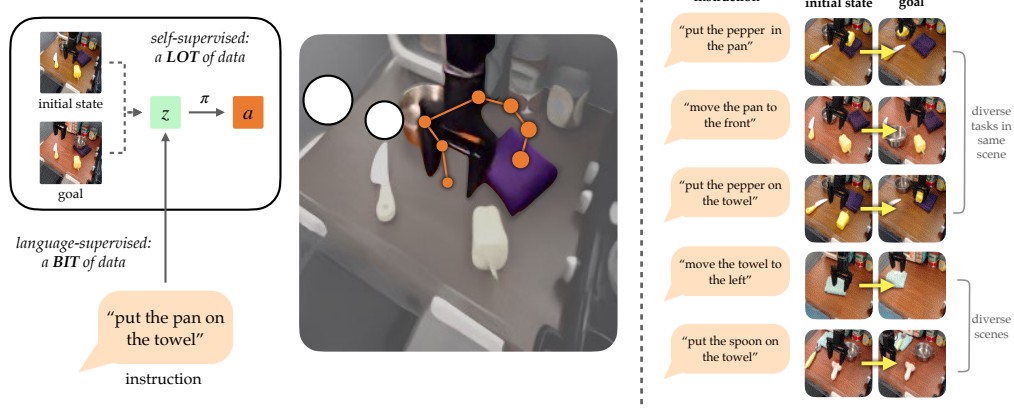

Figure 1: **Left:** Our approach learns representations of instructions that are aligned to transitions from the initial state to the goal. When commanded with instructions, the policy $\pi$ computes the task representation $z$ from the instruction and predicts the action $a$ to solve the task. Our approach is trained with a small number of labeled demonstrations and large-scale unlabeled demonstrations. **Right:** Our approach can solve diverse tasks and generalize to vast environment variations.

Exposing an instruction-following interface for goal-conditioned policies could combine the strengths of both goal- and language- task specification to enable generalist robots that can be easily commanded. While goal-conditioned policy learning can help digest unstructured data, non-robot vision-language data sources make it possible to connect language and visual goals for generalization to diverse instructions in the real world.

To this end, we propose Goal Representations for Instruction Following (GRIF), an approach that jointly trains a language- and a goal- conditioned policy with aligned task representations. We term task representations *aligned* because our objective encourages learning similar representations for language instructions and state transitions that correspond to the same semantic task. GRIF learns this representation structure explicitly through a contrastive task alignment term. Since task representations across language and image goal modalities have similar semantics, this approach allows us to use robot data collected without annotations to improve performance by the agent on image goal tasks when viewed as a goal-conditioned policy, and thus indirectly improve language-conditioned performance in a semi-supervised manner. An overview of GRIF is shown in Figure 1.

We present an approach for learning a language interface for visuomotor control without extensive language labels. With this method, we demonstrate that the semantic knowledge from a pre-trained vision-language model (CLIP [4]) can be used to improved task representations and manipulation even though such models perform poorly at task understanding out-of-the-box. Our experiments show that aligning task representations to scene changes enables improved performance at grounding and following language instructions within diverse real-world scenes.

## 2 Related Work

**Robotic control with language interfaces.** Early works in language-conditioned robotic control use hand-designed parse trees or probabilistic graphical models to convert instructions into symbolic states to configure the downstream planners and controllers [5, 6, 7, 8]. To generalize beyond limited human specifications, a growing number of works have trained conditional policies end-to-end to follow natural language instructions [9, 10, 11, 12, 13, 14, 15, 16, 17, 18, 19, 20]. Combining recent advances large language models (LLMs) [21] with learned language-conditioned policies as a low-level API has have paved the way for broad downstream applications with improved planning and generalization [22, 23, 24, 25, 26, 27]. However, most of these methods need high-capacity policy networks with massive, costly labeled demonstration data. As a result, the learned policies often

generalize poorly to unseen scenarios or can only handle limited instructions in real-world scenes. Unlike past work, we learn low-level language-conditioned control from less annotated data.

**Vision-language pre-training.** Vision-language models (VLMs) enable textual descriptions to be associated with visual scenes [4, 28]. Through contrastive learning over internet-scale data, recent large-scale VLMs such as CLIP [4] have achieved unprecedented zero-shot and few-shot generalization capabilities, with a wide range of applications.

Despite these advances, applying pre-trained VLMs to robotic control is not straightforward since control requires grounding instructions in motions instead of static images. Through training from scratch or fine-tuning on human trajectories [29, 30], recent approaches learn representations for visuomotor control [31, 32]. These works use language labels to to learn visual representations for control without directly using language as an interface to the policy. In CLIPort, Shridhar et al. [33] use pre-trained CLIP [4] to enable sample-efficient policy learning. Their approach selects actions from high-level skills through imitation, assuming access to predefined pick-and-place motion primitives with known camera parameters. In contrast, our approach learns to align the representation of the instruction and the representation of the transition from the initial state to the goal on labeled robot data, and uses these representations for control without assumptions about the observation and action spaces. Other approaches use VLMs to recover reward signals for reinforcement learning [34, 35, 36, 37, 3]. In contrast, our approach directly trains language-conditioned policy through imitation learning without the need for online interactions with the environment.

**Learning language-conditioned tasks by reaching goals.** Alternatively, language-conditioned policies can be constructed or learned through goal-conditioned policies [38, 39]. Lynch and Sermanet [40] propose an approach that facilitates language-conditioned imitation learning by sharing the policy network and aligning the representations of the two conditional tasks. Based on the same motivation, we propose an alternative approach which explicitly extends the alignment of VLMs to specify tasks as changes in the scene. By tuning a contrastive alignment objective, our method is able to exploit the knowledge of VLMs [4] pre-trained on broad data. This explicit alignment improves upon past approaching to connecting images and language [41, 42] by explicitly aligning tasks instead merely jointly training on conditional tasks. In Sec. 5, we show our approach significantly improves the performance of the learned policy and enhances generalization to new instructions.

## 3   Problem Setup

Our objective is to train robots to solve tasks specified by natural language from interactions with the environment. This problem can be formulated as a conditional Markov decision process (MDP) denoted by the tuple $(\mathcal{S}, \mathcal{A}, \rho, P, \mathcal{W}, \gamma)$, with state space $\mathcal{S}$, action space $\mathcal{A}$, initial state probability $\rho$, transition probability $P$, an instruction space $\mathcal{W}$, and discount $\gamma$. Given the instruction $\ell \in \mathcal{W}$, the robot takes action $a_t \in \mathcal{A}$ given the state $s_t$ at each time step $t$ to achieve success.

To solve such tasks, we train a language-conditioned policy $\pi(a|s, \ell)$ on a combination of human demonstrations and autonomously collected trajectories. Since high-quality natural language annotations are expensive and time-consuming to obtain, we assume that only a small portion of the trajectories are labeled with the corresponding instructions. The robot has access to a combination of two datasets—a small-scale labeled dataset $\mathcal{D}_L$ with annotated instructions and a large-scale unlabeled dataset $\mathcal{D}_U$ consists of more diverse play data collected in an open-ended manner. Our goal is to train $\pi(a|s, \ell)$ while taking advantage of both the labeled and unlabeled datasets. We formulate $\pi(a|s, \ell)$ as a stochastic policy that predicts the Gaussian distribution $\mathcal{N}(\mu_a, \Sigma_a)$.

## 4   Goal Representations for Instruction Following

We propose Goal Representations for Instruction Following (GRIF) to interface visuomotor policies with natural language instructions in a semi-supervised fashion (Figure. 2). Although the language-conditioned policy cannot be directly trained on the unlabeled dataset $\mathcal{D}_U$, we can facilitate the training through goal-conditioned tasks. Solving both types of tasks requires the policy to under-

stand the human intent, ground it in the current observation, and predict necessary actions. Although the first steps involve understanding task specifications of different modalities (goal images and language), the remaining steps of such processes can be shared between the two settings. To this end, we decouple the language-conditioned policy $\pi(a|s, \ell)$ into a policy network $\pi_\theta(a|s, z)$ and a language-conditioned task encoder $f_\varphi(\ell)$, where $z = f_\varphi(\ell)$ is the representation of the task specified by the instruction $\ell$. To solve the goal-conditioned task, we also introduce a goal-conditioned task encoder $h_\psi$. The policy network $\pi_\theta$ is shared between the language-conditioned and goal-conditioned tasks.

This approach relies on the alignment of task representations. While most existing VLMs align text with static images, we argue that the representation of the goal-conditioned tasks should be computed from the state-goal pair $(s_0, g)$. This is because the instruction often focuses on the changing factors from the initial state to the goal rather than directly describing the entire goal image, e.g., "*move the metal pan to the left*". Therefore, the representations of goal-conditioned tasks are computed as $z = h_\psi(s_0, g)$ and we aim to train the encoders such that for $(s_0, g, \ell)$ sampled from the same trajectory, the distance between $f_\varphi(\ell)$ and $h_\psi(s_0, g)$ should be close and far apart otherwise. We illustrate our high-level approach in Figure 2.

### 4.1 Explicit Alignment through Contrastive Learning

We propose explicitly aligning the representations of goal-conditioned and language-conditioned tasks through contrastive learning [41]. Compared to implicitly aligning the task presentations through joint training of the two conditional policies, contrastive alignment requires that all relevant information for selecting actions be included in the shared task representation. This improves the transfer between the action prediction tasks for both goal and language modalities by preventing the policy from relying on features only present in one task modality in selecting actions. Using an InfoNCE objective [42], we train the two encoders $f_\varphi$ and $h_\psi$ to represent instructions $\ell$ and transitions $(s_0, g)$ according to their task semantics. More concretely, for $(s_0, g)$ and $\ell$ that correspond to the same task, we would like their embeddings $z_\ell = f_\varphi(\ell)$ and $z_g = h_\psi(s_0, g)$ to be close in the latent space, while $z_\ell$ and $z_g$ corresponding to different tasks to be far apart.

To compute the InfoNCE objective, we define $\mathcal{C}(s, g, \ell) = \cos(f(\ell), h(s, g))$ with the cosine similarity $\cos(\cdot, \cdot)$. We sample positive data $s^+, g^+, \ell^+ \sim \mathcal{D}_L$ by selecting the start state, end state, and language annotation of a random trajectory. We sample negative examples $s^-, g^- \sim \mathcal{D}_L$ by selecting the start state and end state of a random trajectory, and sample $\ell^- \sim \mathcal{D}_L$ by selecting the language annotation of another random trajectory. For each positive tuple, we sample $k$ negative examples and denote them as $\{s_i^-, g_i^-\}_{i=1}^k$ and $\{\ell_i^-\}_{j=1}^k$. Then we can define the InfoNCE $\mathcal{L}_{\text{task}}$:

$$
\begin{aligned}
\mathcal{L}_{\text{lang}\rightarrow\text{goal}} &= -\log \frac{\exp(\mathcal{C}(s^+, g^+, \ell^+)/\tau)}{\exp(\mathcal{C}(s^+, g^+, \ell^+)/\tau) + \sum_{i=1}^k \exp\big(\mathcal{C}(s_i^-, g_i^-, \ell^+)/\tau\big)} \\
\mathcal{L}_{\text{goal}\rightarrow\text{lang}} &= -\log \frac{\exp(\mathcal{C}(s^+, g^+, \ell^+)/\tau)}{\exp(\mathcal{C}(s^+, g^+, \ell^+)/\tau) + \sum_{j=1}^k \exp\big(\mathcal{C}(s^+, g^+, \ell_j^-)/\tau\big)} \\
\mathcal{L}_{\text{task}} &= \mathcal{L}_{\text{lang}\rightarrow\text{goal}} + \mathcal{L}_{\text{goal}\rightarrow\text{lang}}
\end{aligned}
\tag{1}
$$

where $\tau$ is a temperature hyperparameter. $\mathcal{L}_{\text{lang}\rightarrow\text{goal}}$ and $\mathcal{L}_{\text{goal}\rightarrow\text{lang}}$ represent the log classification accuracy of our alignment in predicting goals from language and language from goals respectively.

### 4.2 Weight Initialization with Vision-Language Models

To handle tasks involving objects and instructions beyond those contained in the limited labeled dataset, we wish to incorporate prior knowledge from broader sources into the encoders $f_\varphi$ and $h_\psi$. For this purpose, we investigate practical ways to incorporate Vision-Language Models (VLMs) [4] pre-trained on massive paired images and texts into our encoders. Pre-trained VLMs demonstrate effective zero-shot and few-shot generalization capability for vision-language tasks [4, 43]. However, they are originally designed for aligning a single static image with its caption without the ability to understand the *changes* in the environment that language tasks correspond to, and perform poorly

**Learn Task Representations**      **Learn Policy**

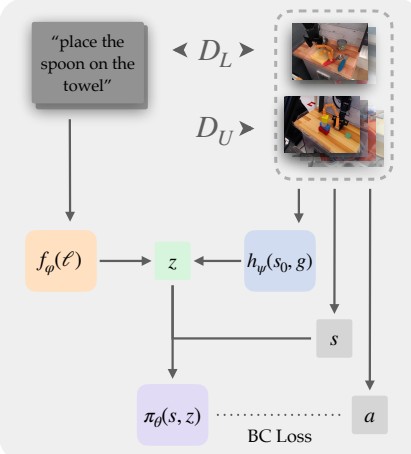

Figure 2: **Left:** We explicitly align representations between goal-conditioned and language-conditioned tasks on the labeled dataset $\mathcal{D}_L$ through contrastive learning. **Right:** Given the pretrained task representations, we train a policy on both labeled and unlabeled datasets.

on compositional generalization [44, 45], which is key to modeling changes in scene state. We wish to encode the change between images while still exploiting prior knowledge in pre-trained VLMs.

To address this issue, we devise a mechanism to accommodate and fine-tune the CLIP [4] model for aligning the transition $(s_0, g)$ with the instruction $\ell$. Specifically, we duplicate and halve the weights of the first layer of the CLIP architecture so it can operate on pairs of stacked images rather than single images. Details on how we modify the pre-trained CLIP to accommodate encoding changes are presented in Appendix C.2. In practice, we find this mechanism significantly improves the generalization capability of the learned policy $\pi_\theta(a|s, g)$.

### 4.3 Policy Learning with Aligned Representations

We train the policy jointly on the two datasets $\mathcal{D}_L$ and $\mathcal{D}_U$ with the aligned task representations. By sampling $(\ell, s_t, a_t)$ from $\mathcal{D}_L$, we train the policy network $\pi_\theta(a|z)$ to solve language-conditioned tasks with $z = f_\varphi(\ell)$. And by sampling $(s_0, g, s_t, a_t)$ from $\mathcal{D}_L \cup \mathcal{D}_U$, $\pi_\theta$ is trained to reach goals with $z = h_\psi(s_0, g)$. We train with behavioral cloning to maximize the likelihood of the actions $a_t$.

We investigate two ways to train the policy given the encoders $f_\varphi$ and $h_\psi$. The straightforward way is to jointly train the policy network $\pi_\phi$ and the two encoders end-to-end. This process adapts the encoders with the policy network to encourage them to incorporate information that facilitates downstream robotic control, but can also backfire if the policy learns to rely on visual-only features that are absent in the language conditioned setting. Alternatively, we can freeze the pre-trained weights of the two encoders and only train the shared policy network $\pi_\phi$ on the two datasets. In Section 5, we evaluate and discuss the performances of both options.

## 5 Experiments

Our work started with the premise of tapping into large, goal-conditioned datasets. To build a language interface for goal-conditioned policy learning, we proposed to learn explicitly aligned task representations, and to align instructions to state changes rather than static goals. Lastly, we advocated for the use of pre-trained VLMs to incorporate larger sources of vision-language knowledge. Therefore, we aim to test the following hypotheses in our experiments:

**H1:** Unlabeled trajectories will benefit the language-conditioned policy on new instructions.

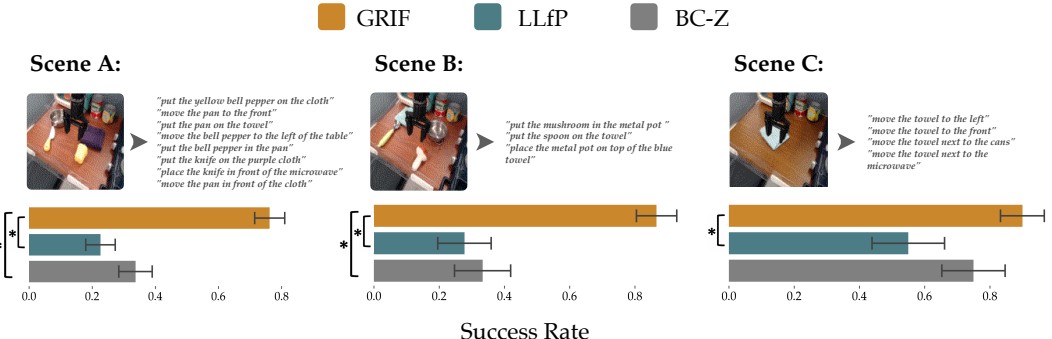

Figure 3: Comparison of success rates ±SE between the top three methods across all trials within the three scenes. Two other baselines LCBC and R3M (not shown) achieved 0.0 success in all evaluation tasks although they do succeed on tasks that are heavily covered in the training data. Statistical significance is starred. The initial observation and instructions of each scene are shown.

**H2:** Explicitly aligning task representations improves upon the implicit alignment from LangLfP-style joint training [40].

**H3:** The prior knowledge in pre-trained VLMs can improve learned task representations.

**H4:** Aligning *transitions* with language enable better use of VLMs compared to conventional image-language contrastive methods [37, 46].

Our experiments are conducted in an table-top manipulation domain. For training, we use a labeled dataset $\mathcal{D}_L$ containing 7k trajectories and an unlabeled $\mathcal{D}_U$ containing 47k trajectories. Our approach learns to imitate the 6 DOF continuous gripper control actions in the data at 5Hz. The evaluation scenes and unseen instructions are shown in Figure 3. Additional details about the environment, the dataset, and the breakdown of results are described in Appendices B and E.

## 5.1 Comparative Results

We compare the proposed GRIF with four baseline methods on a set of 15 unseen instructions from all 3 scenes and report the aggregated results in Figure 3, with GRIF attaining the best performance across all scenes. The per-task success rates can be found in Appendix E. **LCBC** [9] uses a behavioral cloning objective to train a policy conditioned on language from $\mathcal{D}_L$, similar to prior methods on instruction-conditioned imitation learning. **LLfP** [40] jointly trains a goal conditioned and language conditioned policy on partially labeled data, but does not learn aligned task representations. **R3M** [32] provides pre-trained state representations for robot manipulation that are predictive of language-conditioned rewards. We use this approach as a baseline by jointly training goal- and language-conditioned policies while using R3M state encodings as goal representations (i.e., $h_\psi(s_0, g) = \text{R3M}(g)$). **BC-Z** [10] jointly trains language- and video-conditioned policies and uses an additional cosine similarity term to align video and language embeddings. This approach does not transfer directly into our goal-conditioned setting, but we create a baseline that adapts it to our setting by jointly training goal- and language-conditioned policies while aligning task representations with a cosine distance loss. The architecture choices are standardized across all methods for fair comparisons. Unless stated otherwise, all baselines use a ResNet-18 as the goal encoder $h_\psi(s_0, g)$. In our preliminary experiments, this architecture was found to give good performance when used to train goal-conditioned policies in our setting. For the language encoder $f_\varphi(\ell)$, all baselines use a pre-trained and frozen MUSE model [47], as in previous work [40, 10].

We find that language-conditioned policies must make use of unlabeled trajectories to achieve non-zero success rates when generalizing to new language instructions in support of **H1**. LCBC does not use unlabeled data and fails to complete any tasks. R3M jointly trains goal- and language-conditioned policies, but it also fails all tasks. This is likely due to its goal encodings being frozen and unable to be implicitly aligned to language during joint training. Methods that use implicit or explicit alignment (GRIF, LLfP, BC-Z), are able to exploit unlabeled goal data to follow instruc-

tions to varying degrees of success. These comparisons suggest that the combined effect of using pre-trained CLIP to align transitions with language significantly improves language-conditioned capabilities. Our model significantly outperformed all baselines on 8 out of 15 tasks, achieving high success rates on several tasks where the baselines almost completely fail (*"place the knife in front of the microwave", "move the pan in front of the cloth", "put the knife on the purple cloth"*), while achieving similar performance to the next-best baseline on the remaining tasks. Where baselines failed, we often observed *grounding* failures. The robot reached for incorrect objects, placed them in incorrect locations, or was easily distracted by nearby objects into performing a different task.

## 5.2 Ablation Study

We run a series of ablations to analyze the performance of GRIF and test the hypotheses. **No Align** ablates the effect of explicit alignment by removing the contrastive objective. We also unfreeze the task encoders so that they are implicitly aligned via joint training of the language- and goal-conditioned policies. **No CLIP** ablates the effect of using pre-trained CLIP by replacing the image and text encoders with a ResNet-18 and pre-trained MUSE language encoder. In **No Start**, the image task representaions only depend on goals as $h_\psi(g)$, instead of depending on transitions as $h_\psi(s_0, g)$. This is the conventional way to connect goals and language with CLIP that is often used in previous work [46, 37]. For **GRIF (Labeled)**, we exclude $\mathcal{D}_U$ to study whether using unlabeled data is important for performance. **GRIF (Joint)** trains the task alignment and policy losses jointly, taking gradients through the image encoder and freezing the language encoder. This is the end-to-end approach discussed in Section 4.3. We refer to our full approach without joint training as **GRIF (Frozen)** in the remainder of this section.

As shown in Figure 4, explicit alignment, pre-trained CLIP, and transition-based task representations all play critical roles in achieving high success rates. Notably, the conventional approach of aligning static goals and instructions with CLIP (**No Start**) fails almost completely in our setting. This is in support of **H4** and confirms that transitions, and not goal images, should be aligned to language tasks. In **GRIF (Labeled)**, dropping $\mathcal{D}_U$ significantly decreases

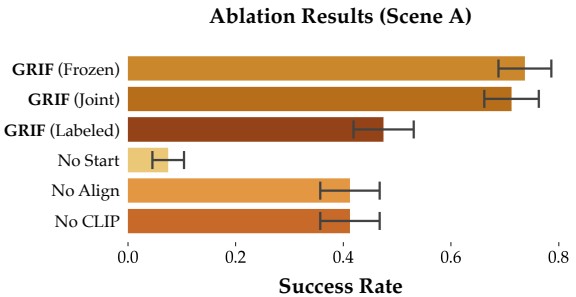

Figure 4: Success rates of ablations with one standard error.

success rates, further supporting **H1**. We observe that this is primarily due to a deterioration of manipulation skills rather than grounding, which is expected as grounding is mostly learned via explicit alignment on $\mathcal{D}_L$. Regarding **H2** and **H3**, we observe that removing either alignment or CLIP results in a large drop in performance. We also observed that **No Align** outperforms its counterpart *LLfP* by using the pre-trained CLIP model (after the modification in Sec. 4.2) in the task encoder. We hypothesize that this is because CLIP has already been explicitly aligned during pre-training, and some of its knowledge is retained during joint training with the policy even without GRIF's task alignment loss. Lastly, deciding to freeze the task encoders during policy training does not appear to significantly affect our model's performance. This is likely because the contrastive learning phase already learns representations that can represent the semantic task, so there is less to gain from further implicit alignment during joint training.

## 5.3 Analysis on the Learned Task Representations

For additional analysis, we evaluate our model's task grounding capabilities independently of the manipulation policy and compare it with ablations. Specifically, we evaluate how well our model can connect new language instructions to correct goals in a scene. This is important to downstream policy success: if the model is able to project the language to a representation $f_\varphi(l)$ that is close to

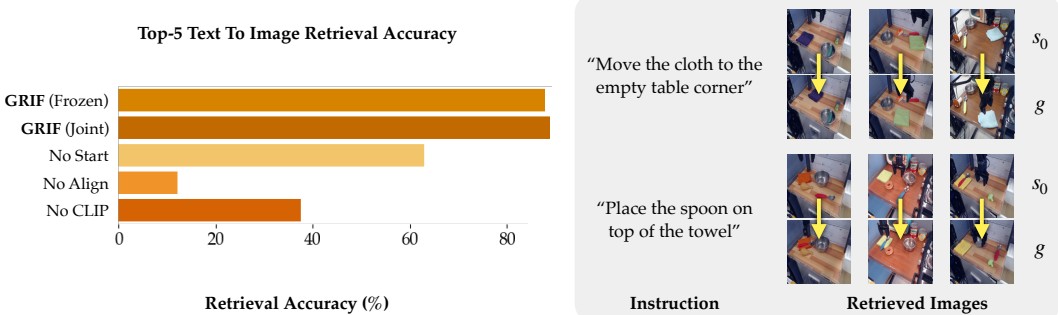

Figure 5: Left: Comparison of the top-5 text to image retrieval accuracy of representations learned by different ablations. Right: Examples of retrieved image pairs given instructions.

that of the correct (but unprovided) goal $h_\psi(s_0, g)$, then the policy will likely be able to execute the task since it has been trained on a large amount of goal-conditioned data.

Our task representations are trained with a contrastive objective, offering a convenient way to compute alignment between language and goals. On an dataset of labeled held-out trajectories, we compute the similarities between all pairs of visual task representations $h_\psi(s_0, g)$ and language task representations $f_\varphi(\ell)$. For each language instruction, we retrieve the top $k$ most similar $(s_0, g)$ transitions and compute the accuracy for the correct transition being retrieved. We compute this metric in fixed batches of 256 examples and average over the validation set to report a text-to-image retrieval accuracy. We compute this metric for representations from each of our ablations and report the results in Figure 5 to analyze why GRIF outperforms other approaches in our main experiments. Our task representations show significantly better generalization compared to using a conventional image-language alignment (**No Start**), despite it being CLIP's original pre-training objective. The alignment accuracy is also more than 50% higher than when using non-VLM encoders (**No CLIP**), suggesting potentially large gains in grounding capability through using VLMs.

We also study the effect of the number of language annotations on our model's grounding capability. Even at less than half the number of language annotations (3k), GRIF outperforms all the ablations in Figure 5, achieving a retrieval accuracy of 73%. Detailed results for this ablation are presented in Appendix F, showing our approach is robust to lower amounts of language supervision.

## 6    Discussion, Limitations, and Future Work

Our approach to aligning image goals and language instructions enables a robot to utilize large amounts of unlabeled trajectory data to learn goal-conditioned policies, while providing a "language interface" to these policies via aligned language-goal representations. In contrast to prior language-image alignment methods, our representations align *changes* in state to language, which we show leads to significantly better performance than more commonly used CLIP-style image-language alignment objectives. Our experiments demonstrate that our approach can effectively leverage unlabeled robotic trajectories, with large improvements in performance over baselines and methods that only use the language-annotated data.

**Limitations and future work.** Our method has a number of limitations that could be addressed in future. For instance, our method is not well-suited for tasks where instructions say more about *how* to do the task rather than *what* to do (e.g., "*pour the water slowly*")—such qualitative instructions might require other types of alignment losses that more effectively consider the intermediate steps of task execution. Our approach also assumes that all language grounding comes from the portion of our dataset that is fully annotated or a pre-trained VLM. An exciting direction for future work would be to extend our alignment loss to utilize non-robot vision-language data, such as videos, to learn rich semantics from Internet-scale data. Such an approach could then use this data to improve grounding on language not in the robot dataset and enable broadly generalizable and powerful robotic policies that can follow user instructions.

**Acknowledgements**

We would like to acknowledge the funding provided by AFOSR FA9550-22-1-0273, ONR N00014-20-1-2383, NSF IIS-2150826, and ONR YIP N00014-20-1-2736. The COCOSYS SRC center also contributed funding for post-submission updates.

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

# Appendix

## A  Website

Videos and code for our approach can be found at https://rail-berkeley.github.io/grif/.

## B  Environment Details

We provide more details on the real-world environment in this section.

### B.1  Robot

We use a 6DOF WidowX 250 robot with a 1DOF parallel-jaw gripper. We install the robot on a tabletop where it can reach and manipulate objects within an environment set up in front of it. The robot receives inputs from a Logitech C920 RGB camera installed in an over-the-shoulder view. The images are passed into the policy at a 128 x 128, and the control frequency is 5Hz. Teleoperation data is collected with a Meta Quest 2 VR headset that controls the robot.

### B.2  Dataset Details

The dataset consists of trajectories collected from 24 different environments, which includes kitchen-, sink-, and tabletop-themed manipulation environments. The dataset features around 100 objects, including containers, utensils, toy food items, towels, and other kitchen-themed objects. It includes demonstrations of 13 high-level skills (pick and place, sweep, etc.) applied to different objects. Out of the 54k total trajectories, 7k are annotated with language instructions. Around 44k of the trajectories are expert demonstrations and around 10k are collected by a scripted policy.

## C  Method Details

### C.1  Policy Network

Our policy network $\pi_\theta(a|s, z)$ uses a ResNet-34 architecture. To condition on the task embedding $z$, it is first passed through 2 fully connected layers. Then, the policy network is conditioned on the embedding using FiLM layers, which are applied at the end of every block throughout the ResNet. The image encoding is then passed into a fully connected network to predict the action distribution. The policy network predicts the action mean, and we use a fixed standard deviation.

### C.2  CLIP Model Surgery

Instead of separately encoding $s_0$ and $g$ inside $f_\varphi$, we perform a "surgery" to the CLIP model to enable it to take $(s_0, g)$ as inputs while keeping most of its pre-trained network weights as intact as possible. Specifically, we clone the weight matrix $W_{\text{in}}$ of the first layer in the pre-trained CLIP and concatenate them along the channel dimension to be $[W_{\text{in}}; W_{\text{in}}]$, creating a model that can accept the stacked $[s_0, g]$ as inputs. We also halve the values of this new weight matrix to make it $W'_{\text{in}} = [W_{\text{in}}/2; W_{\text{in}}/2]$, ensuring its output $0.5(W_{\text{in}}s_0 + W_{\text{in}}g)$ will follow a distribution similar to the output by the original first layer $W_{\text{in}}s_0$. While this surgery alone cannot perfectly close the gap, the resultant modified encoder can serve as a capable initialization for the transition encoder $h_\psi$. We further fine-tune $h_\psi$ on the labeled robot dataset $\mathcal{D}_L$ using the aforementioned method to adapt it for instruction-following tasks.

### C.3  Negative Sampling

For training the contrastive objective on $\mathcal{D}_L$, our batch sampling strategy is non-standard. We use 2 dataloaders in parallel; the first samples from shuffled trajectories, while the second iterates through

trajectories in the order that they are stored in the dataset. Each samples batches of 128 trajectories and they are concatenated to produce a batch size of 256. The reason for this is that if we were to use a standard sampling strategy, most examples in a batch would be from different scenes. This is not useful for the contrastive loss because the representations would just learn to distinguish tasks based on the set of objects that appear. The robot benefits from being able to distinguish different tasks in the same scene, so we try to include many trajectories from the same scene in each batch. Using an unshuffled dataloder is a convenient way to achieve this since trajectories from the same scene are stored together. This can be considered a form of negative mining for the contrastive learning stage.

### C.4 Instruction Augmentation

In order to increase the diversity of language annotations, we augment our natural language annotations using GPT-3.5. Through the API, we query the gpt-3.5-turbo model to generate paraphrases of instructions in our dataset. We generate 5 paraphrases per instruction and sample from them randomly during training. An example prompt and response are shown below. We found in preliminary experiments that using augmented instructions slightly improved language generalization, so we keep this augmentation for all models and baselines.

```
Prompt:
    Generate 5 variations of the following command:
    "put the mushroom in the metal pot"
    Number them like 1. 2. 3.
    Be concise and use synonyms.
Response:
    1. Place the fungus in the metallic container.
    2. Insert the mushroom into the steel vessel.
    3. Set the toadstool inside the iron cauldron.
    4. Position the champignon within the tin pot.
    5. Place the fungi in the metallic kettle.
```

### C.5 Goal Relabeling

For unlabeled trajectories in $\mathcal{D}_U$, we use a simple goal relabeling strategy: with 50% probability, we use the final achieved state as the goal, and with 50 % probability we uniformly sample an intermediate state in the trajectory to use as the goal. We do not relabel the annotated trajectories in $\mathcal{D}_L$.

### C.6 Hyperparameters

When training the task encoders using the contrastive learning objective, we use a batch size of 256. We reduce the batch size to 128 when we train the policy network. We use the Adam optimizer with a learning rate schedule that uses linear warmup and cosine decay. The peak learning rate is 3e-4 for all parameters except the CLIP ViT encoders, for which we use 3e-5. We use 2000 warmup steps and 2e6 decay steps for the learning rate schedule. When we jointly train the alignment and behavioral cloning losses, we use a weight of 1.0 on both terms. These hyperparameters were found through random search. We train our models for 150k steps, which takes around 13 hours on 2 Google Cloud TPU cores.

## D  Experimental Details

The scenes were constructed with the objects shown in Table 1 within a toy kitchen setup.

During evaluation, we roll out the policy given the instruction for 60 steps. Task success determined by a human according to the following criteria:

- Tasks that involve putting an object into or on top of a container (e.g. pot, pan, towel) are judged successes if any part of the object lies within or on top of the container.

- Tasks that involve moving an object toward a certain direction are judged successes if the object is moved sufficiently in the correct direction to be visually noticeable.

- Tasks that involve moving an object to a location relative to another object are judged successes if the object ends in the correct quadrant and are aligned with the reference object as instructed. For example, in "place the knife in front of the microwave", the knife should be placed in the top-left quadrant, and be overlapping with the microwave in the horizontal axis.

- If the robot attempts to grasp any object other than the one instructed, and this results in a movement of the object, then the episode is judged a failure.

Table 1: Evaluation Scenes

| Scene | Objects |
|-------|---------|
| A | knife, pepper, towel, & pot |
| B | mushroom, towel, spoon, & pot |
| C | towel |

# E   Experimental Results

We show per-task success rates for our approaches, the baselines, and the ablations in this section. The tasks in scenes A and B were evaluated for 10 trials each, while those in C were evaluated for 5 trials.

Table 2: Comparison of Approaches

| Scene | Task | Success Rate | | | | |
|-------|------|------|------|------|------|------|
| | | **GRIF** | **LCBC** | **LLfP** | **R3M** | **BC-Z** |
| A | put the yellow bell pepper on the cloth | **0.6** | 0.0 | 0.0 | 0.0 | **0.6** |
| | move the pan to the front | **1.0** | 0.0 | 0.6 | 0.0 | 0.0 |
| | put the pan on the towel | 0.8 | 0.0 | 0.3 | 0.0 | **0.9** |
| | move the bell pepper to the left of the table | 0.7 | 0.0 | 0.0 | 0.0 | **0.8** |
| | put the bell pepper in the pan | **0.8** | 0.0 | 0.1 | 0.0 | 0.3 |
| | put the knife on the purple cloth | **0.7** | 0.0 | 0.2 | 0.0 | 0.0 |
| | place the knife in front of the microwave | **0.7** | 0.0 | 0.0 | 0.0 | 0.1 |
| | move the pan in front of the cloth | **0.6** | 0.0 | 0.3 | 0.0 | 0.0 |
| B | put the mushroom in the metal pot | **0.9** | 0.0 | 0.5 | 0.0 | 0.4 |
| | put the spoon on the towel | **0.9** | 0.0 | 0.3 | 0.0 | 0.4 |
| | place the metal pot on top of the blue towel | **0.8** | 0.0 | 0.0 | 0.0 | 0.2 |
| C | move the towel to the left | **1.0** | 0.0 | **1.0** | 0.0 | **1.0** |
| | move the towel to the front | **1.0** | 0.0 | **1.0** | 0.0 | **1.0** |
| | move the towel next to the cans | **0.6** | 0.0 | 0.0 | 0.0 | 0.2 |
| | move the towel next to the microwave | **1.0** | 0.0 | 0.2 | 0.0 | 0.8 |

Table 3: Comparison of Ablations

| Scene | Task | Success Rate | | |
|---|---|---|---|---|
| | | **GRIF (Frozen)** | **GRIF (Joint)** | **GRIF (Labeled)** |
| A | put the yellow bell pepper on the cloth | 0.6 | 0.8 | **1.0** |
| | move the pan to the front | **1.0** | **1.0** | 0.7 |
| | put the pan on the towel | 0.8 | **1.0** | 0.1 |
| | move the bell pepper to the left of the table | **0.7** | 0.4 | 0.2 |
| | put the bell pepper in the pan | 0.8 | 0.6 | **1.0** |
| | put the knife on the purple cloth | **0.7** | 0.4 | 0.1 |
| | place the knife in front of the microwave | **0.7** | 0.6 | 0.5 |
| | move the pan in front of the cloth | 0.6 | **0.9** | 0.2 |
| | | **No Start** | **No Align** | **No CLIP** |
| A | put the yellow bell pepper on the cloth | 0.3 | 0.5 | 0.0 |
| | move the pan to the front | 0.6 | 0.8 | 0.0 |
| | put the pan on the towel | 0.6 | 0.6 | 0.0 |
| | move the bell pepper to the left of the table | 0.4 | 0.6 | 0.2 |
| | put the bell pepper in the pan | 0.7 | 0.6 | 0.1 |
| | put the knife on the purple cloth | 0.2 | 0.2 | 0.0 |
| | place the knife in front of the microwave | 0.1 | 0.0 | 0.0 |
| | move the pan in front of the cloth | 0.4 | 0.0 | 0.3 |

# F    Ablation of Number of Annotations

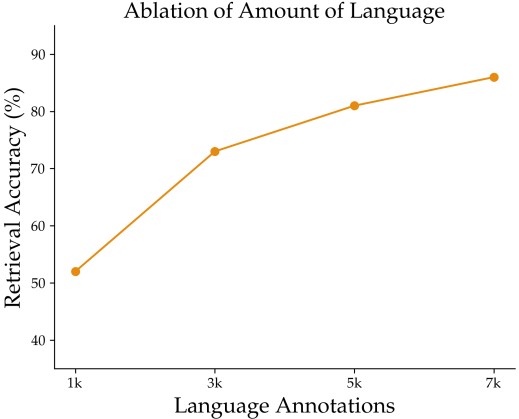

Figure 6: Scaling of GRIF grounding capability by number of language annotations available.

We ablate effect of the amount of language supervision on GRIF's grounding capabilities. We compute the (top-5) text-to-image retrieval accuracy of GRIF representations when trained on 7k, 5k, 3k, and 1k annotations, and find accuracies of 86%, 81%, 73%, and 52% respectively. These accuracies are plotted in Figure 6. By comparing these accuracies with the grounding performance of the ablations in Figure 5, we see GRIF enables more robust grounding with little language supervision.

