# OpenReview forum: "Goal Representations for Instruction Following: A Semi-Supervised Language Interface to Control"
_robot-learning.org/CoRL/2023/Conference — CoRL 2023 Poster_

### Official Review · Reviewer_HoyX · 2023-07-18

**Confidence:** 4
**Originality:** Very Good
**Technical Quality:** Excellent
**Clarity Of Presentation:** Very Good
**Impact:** 3

**Recommendation:**

Strong Accept: I recommend accepting the paper and will argue for my recommendation even if other reviewers hold a different opinion.

**Review:**

## Strengths
1. **Positive results with thorough ablations**: The results of the method are strong even against relevant, similarly powerful baselines. The authors ablate all of the components of their model and demonstrate large performance changes, motivating their approach.

2. **Clear presentation**: The paper is clearly-written and the problem is well-motivated. The description of the system is generally easy to follow and the hypotheses tested are expressed explicitly.

3. **Makes use of existing data**: As the authors point out, collecting language data is expensive and difficult, while image-based goals can be easily collected from hindsight relabelling or similar methods.

## Weakenesses
1. **Limited language**: The language used seems to be from fairly basic templates. This somewhat invalidates the motivation of the paper, which is that collecting NL data is expensive -- collecting data from templates is free. The expensive bit is collecting data from real people. While CLIP might generalize to more complex language, the results here do not convincingly show that the method can handle linguistic variation.

2. **Short-horizon/simple tasks**: As far as I can tell, the tasks considered are relatively short-horizon (single pick/place). Adding the initial state $s_0$ here might have an outsized impact, as compared to a longer-horizon task where there might be a greater difference between states. The authors do address this somewhat in their limitations.


**Quality Of The Limitations Section:**

Limitations are addressed clearly

**Questions For Rebuttal:**

1. CLIP has been shown in the past to struggle with compositional generalization (https://arxiv.org/pdf/2212.10537.pdf, https://arxiv.org/pdf/2211.13854.pdf, i.a.). Right now, the paper does not explicitly test whether the CLIP-based model can perform equally well on instructions requiring a compositional analysis of the input. For example, are the instructions "Place the pot on the towel" and "place the towel on the pot" both handled successfully? More generally, I'm curious as to whether the explicit image alignment (and the inclusion of the initial state) helps in these cases. It seems like it should.

2. What is the actual form of $\pi$? Is it an MLP? How many layers, etc. etc.

3. How does the approach scale w.r.t. language data? The paper uses 7k demonstrations, but what is the curve here, e.g. at 1k, 3k, 5k, etc.. What is the minimum number of language annotations you can get away with?

4. On L109 the unlabelled dataset is $D_U$ but later on L116 it's $D_B$. Is this a typo?

**Robotics Focus:**

Sufficient demonstration on hardware

**Summary Of Paper:**

The paper points out that the cost of obtaining goals expressed visually is much lower than the cost of obtaining language goals, but that language goals are often easier for human operators to provide. Accordingly, the authors propose to use a small amount of language instructions paired with a large amount of goals for learning manipulation policies. The method projects language and goals into a shared space, where the representations of each modality are aligned using a small amount of labelled data. The goals are additionally conditioned on a start state, meaning the goal representation encodes a difference from the start state rather than just a goal. Ablations show both the alignment and start state are important to performance. The authors demonstrate their method on hardware.

**Summary Of Recommendation:**

Based on the paper's strengths, I'm recommending a strong accept. The weaknesses I described are fundamental shortcomings of any approach to the tasks tackled here, and so I think they are outside of the scope of what could be tackled in a rebuttal period, and they would hold of any other paper in the area as well. The method proposed seems like it is a useful method for addressing the state of the world as it is now (image goals being more available than text goals, but less usable). This may change in the future, but for the time being, the method given in the paper fills an existing gap.

---

### Official Review · Reviewer_mB1L · 2023-07-21

**Confidence:** 5
**Originality:** Good
**Technical Quality:** Fair
**Clarity Of Presentation:** Good
**Impact:** 3

**Recommendation:**

Weak Reject: I recommend rejecting the paper, but will not argue for my recommendation if the majority of other reviewers have a different opinion.

**Review:**

STRENGTHS
+ The ability to train a language understanding model with relatively few labeled datapoints and a larger set of unlabeled datapoints is of practical importance.
+ The use of pre-trained vision-language models is sensible (and standard), though there are questions about the means by which they are adapted as noted below.
+ The idea of coupling the initial and goal states together when learning the task embedding is sensible, particularly for tasks that can not be specified in terms of the goal state alone (i.e., as a means of learning a representation of the task as the "change" in the state).
+ With the exception of a small number of grammatical errors, the paper is well written and well organized.



WEAKNESSES
- The overall method is only evaluated on 15 different commands that involve pick-and-place operations with a smaller number of objects (i.e., many of the instructions differ only in the pick and place object/location). Of these, the proposed model performs better than the baselines on 8 instructions. Given the small sample size, it is hard to tell whether these gains are significant. Note that under "Robotics Focus" below, there is no option for "some hardware experiments but more necessary" and so I selected "sufficient demonstration on hardware" over the other options, but I feel that more experiments are needed.
- The paper neither cites nor compares to CLIPort, which has been shown to be capable of pick-and-place tasks very similar to those considered here, at least those that don't involve relative motion (i.e., Scenes A and B, which involve 11 out of 15 of the instructions). The paper should provide a quantitative comparison to CLIPort along with a discussion of their differences.
- The main paper provides no details regarding how it adapts and fine-tunes CLIP to model changes in state. Given that the paper suggests that this is a contribution that significantly improves the generalization of the framework, and given the questions about the ablations below, it is important that the details be provided in the main paper.
- Out of the 15 instructions, only 4 (perhaps only 2) (Scene C) require reasoning over the relative location of an object. It is not clear why explicitly representing the task in terms of the initial and final states is necessary. Interestingly, the performance gap between GRIF and the baselines (Fig. 3) is the smallest for these 4 instructions compared to those in Scenes A and B.
- Related, it is not obvious why the performance of the "No Start" ablation is so poor given that the task doesn't require reasoning over the change between the start and goal states. Without more detail regarding the way in which CLIP was modified, one wonders if this is a result of these modifications (unless a vanilla CLIP was used for this ablation vs. the modified CLIP but without the initial state).
- Unless I missed it, there is no discussion in the main paper or appendix of the robot's action space.
- The contributions of the analysis of the learned task representations (Section 5.3) is limited. Further, as the method is using a pre-trained CLIP, the extent to which the language is "new" is unclear.
- Referring to the fact that the model reasons over the initial and goal states as involving "trajectories" or involves understanding "dynamics or motions" is a bit of an over-sell. Start and goal states hardly constitute trajectories or dynamics.



MINOR:
* Lines 24-25: Stating that the model must ground intent in the current observation is an over-simplification. More generally, it requires grounding language in the context of a representation of the robot's action and state spaces, the later of which includes more than the current image.
* Line 27: While neural network models require large amounts of data, the same is not true of traditional structured approaches that use learning, albeit at the expense of significant feature engineering.
* Section 2: It is surprising that the paper makes no mention of LLM-based approaches to language understanding for robots.
* Section 2: The paper is missing earlier work in imitation learning for language understanding [1,2,3]

GRAMMAR:
* Line 70: "apply pre-trained VLMS for robotic control" &rarr; "**applying** pre-trained VLMS **to** robotic control"
* Line 159: "capability" &rarr; "capabilities"
* Line 183: "we proposed to learn explicitly aligned"?




REFERENCES

[1] H. Mei, M. Bansal, and M. Walter. Listen, attend, and walk: Neural mapping of navigational instructions to action sequences. In Proceedings of the National Conference on Artificial Intelligence (AAAI), 2016.

[2] P. Anderson, Q. Wu, D. Teney, J. Bruce, M. Johnson, N. Sünderhauf, I. D. Reid, S. Gould, and A. van den Hengel. Vision-and-language navigation: Interpreting visually-grounded navigation instructions in real environments. In Proceedings of the IEEE/CVF Conference on Computer Vision and Pattern Recognition (CVPR), 2017.

[3] D. Fried, R. Hu, V. Cirik, A. Rohrbach, J. Andreas, L.-P. Morency, T. Berg-Kirkpatrick, K. Saenko, D. Klein, and T. Darrell. Speaker-follower models for vision-and-language naviga- tion. In Advances in Neural Information Processing Systems (NeurIPS), Dec. 2018.

[4] M. Shridhar, L. Manuelli, and D. Fox. CLIPort: What and Where Pathways for Robotic Manipulation. In Proceedings of the Conference on Robot Learning (CoRL), 2021.



**Quality Of The Limitations Section:**

Limitations are addressed clearly

**Questions For Rebuttal:**

1) What is the nature of the acton space?
2) As noted above, can the authors speak to the differences relative to CLIPort and (ideally) provide a quantitative comparison?

**Robotics Focus:**

Sufficient demonstration on hardware

**Summary Of Paper:**

The paper describes a language understanding architecture for pick-and-place-like tasks that can be trained using only a relatively small amount of annotated data paired with a larger amount of unlabeled (self-play) data. Integral to the framework is the idea of learning encoders that align the task representation expressed in natural language $l$ with an encoding of the corresponding start-goal pair $(s_0,g)$. The paper learns this alignment using contrastive learning based on $(s_0,g,l)$ tuples from the labeled dataset. The unlabeled dataset (i.e., $(s_0,g)$, but no $l$) is used to fine-tune a pretrained vision-language model (CLIP). The method trains a downstream policy that is conditioned on the task embedding either jointly or with the embeddings frozen. The paper compares the method to several recent baselines on a series of table-top pick-and place tasks and ablates the different model components.

**Summary Of Recommendation:**

The paper considers an important problem and the topic of language understanding for robotics is of significant interest lately. The paper is missing qualitative and (ideally) quantitative comparisons to a highly relevant baseline. Some of the results are counter to what is expected, particularly those that involve ablations of the initial state. Further, the analysis is limited to only a handful of very similar instructions. Together, this makes it difficult to evaluate the significance of the contributions.

---

### Official Review · Reviewer_z7wQ · 2023-07-23

**Confidence:** 4
**Originality:** Good
**Technical Quality:** Very Good
**Clarity Of Presentation:** Excellent
**Impact:** 3

**Recommendation:**

Strong Accept: I recommend accepting the paper and will argue for my recommendation even if other reviewers hold a different opinion.

**Review:**

Strengths
- The proposed method enables robots to get a language interface with little data. This is useful for training robots when the labeled data is limited.
- This paper clearly presents the hypotheses of the proposed method and provides the experiments to verify them.
- The paper presents the method and results clearly. It is easy to follow the paper.

Weaknesses
- The idea to use contrastive learning to leverage other modalities (where more unlabeled data is available) in robot control has been explored in some of the prior work. For example, Language-Image Value learning is a recent work that leverages video data and contrastive learning for learning a reward for robot control. The paper needs to discuss how the proposed method differs from the prior explorations to demonstrate the novelty.
- While the GRIF outperforms all baselines, it is worth noting that the baseline methods employ ResNet18 as the image encoder whereas GRIF employs a modified CLIP model which uses ResNet50 as the image encoder. Some of the performance improvement may come from the bigger model capacity of ResNet50. It will be a fairer comparison if all presented methods can use the same ResNet50 encoder as in the modified CLIP.
- The generalization evaluation is limited. They use a retrieval task (i.e. retrieval of transitions specified in natural language) to demonstrate that the learned representation can generalize to other language instructions. However, this is demonstrated in the experiments for the original CLIP, and it doesn’t show if this representation can help generalization in learning a language-conditioned policy. A better experiment will be having a test set where the language instructions are novel and measure the robot success rate of the novel instructions.

Ma, Y. J., Liang, W., Som, V., Kumar, V., Zhang, A., Bastani, O., & Jayaraman, D. (2023). LIV: Language-Image Representations and Rewards for Robotic Control. ICML 2023.


**Quality Of The Limitations Section:**

Limitations are addressed clearly

**Questions For Rebuttal:**

- As discussed in the weaknesses, the paper needs to address the questions about novelty, use of image encoder in the experiment, and the test for generalization
- This paper claims that GRIF can be trained with fewer language annotations. The appendix shows that the totral number of trajectories is 54k and 7k of them contain language annotation. The appendix also mentioned that the language instructions are augmented using ChatGPT3.5 so this brings the language instruction to 35k. In this case, the number of language instructions is not as little as one may imagine. How does data augmentation impacts the performance of GRIF? And how sensitive the proposed performance is to the number of annotated trajectories? These experiments will be important to demonstrate the effectiveness of the proposed method.
- (Minor) The tested tasks seem relatively simple, consisting only of "put," "place," and "move" instructions. If more complex tasks (like in other datasets, e.g. CALVIN, MetaWorld, etc) were included, such as "slide the door to the left", "rotate the red block", or “turn faucet off,” how will this impact the proposed method?


**Robotics Focus:**

Sufficient demonstration on hardware

**Summary Of Paper:**

This paper proposes GRIF, an approach that leverages unlabeled trajectory data to train a language-conditioned policy. GRIF first train a shared task representation between language and task transition (i.e. pair of the initial state and the goal) to align both embeddings. Then they can train the goal-conditioned and language-conditioned policy together and leverage the unlabeled goal-conditioned data to improve the language-conditioned policy. This allows the robot to learn a language interface with few labeled language data. The experiments on real robots show that the learned policy outperforms baseline language-conditioned policy and the aligned representation has a better generalization in a retrieval task.

**Summary Of Recommendation:**

The paper is well presented and the proposed method can potentially benefit training a policy with few labeled data. However, I’m leaning toward rejection, as discussed in the weaknesses and questions, it is unclear about the novelty, generalization, and how the language augmentation impacts the presented method.

-----

Post rebuttal update

The authors' responses addressed my questions. While there are limitations and shortcomings as pointed out by reviewer HoyX, it is a useful approach to leveraging visual goals to improve the limited language goal data and a well-executed system, I'm updating my recommendation to acceptance.

---

### Author Response · Authors · 2023-08-10
**General Response**

We thank the reviewers for their thoughtful and detailed feedback. We are glad that the reviewers thought our approach tackled an important problem and found our presentation to be clear. We have addressed the questions and concerns of the reviewers in our responses to each of them, and revised our manuscript in red based on their suggestions. Below we summarize the key concerns we have addressed.

**Summary:**

- We  provide further discussion of the novelty of GRIF compared to LIV and CLIPort. We also explain the different assumptions and problem formulations in these works, which prevent them from being directly applied as baseline methods in our experiments.
- Through additional ablation experiments, we investigate how sensitive GRIF and the baselines are to details like encoder architecture and data augmentations.
- We clarify that our main experiments show GRIF significantly outperforms all baselines across all scenes tested in the real-world when using language instructions that were unseen during training.
- We run an additional ablation experiment demonstrating that our approach scales well with the amount of language annotations.

---

### Decision · Program_Chairs · 2023-08-30

**Decision:**

Accept (Poster)

**Comment:**

This paper proposes a  method that explores joint image- and goal- conditioned policies with language using only a small amount of language data. All the reviewers find the proposed problem approach of practical importance. The authors have provided careful answers to the concerns of the reviewers with additional experiments, in particular in terms of novelties with some state of the art baselines. The submission receives one accept, one strong accept and one weal reject but the reviewer does not argue his recommendation.